# Impact of integrative care on cardiovascular disease risk in newly diagnosed type 2 diabetes mellitus patients: A BI-VitalLife Cohort study

**Tanawat Khunlertkit[1], Teeradache Viangteeravat[1], Panupong Wangprapa[1], Suthee Siriwechdaruk[1,2], Jeremy Mark Ford[1], Krit Pongpirul**[1,3,4,5]*

1 Bumrungrad International Hospital, Bangkok, Thailand, 2 VitalLife Scientific Wellness Center, Bangkok, Thailand, 3 Center of Excellence in Preventive and Integrative Medicine and Department of Preventive and Social Medicine, Faculty of Medicine, Chulalongkorn University, Bangkok, Thailand, 4 Department of International Health, Johns Hopkins Bloomberg School of Public Health, Baltimore, Maryland, United States of America, 5 Department of Infection Biology & Microbiomes, Faculty of Health and Life Sciences, University of Liverpool, Liverpool, United Kingdom

* doctorkrit@gmail.com, krit.po@chula.ac.th, kpongpi1@jhu.edu, pongpiru@liverpool.ac.uk, kritp@bumrungrad.com

**Data Availability Statement:** All relevant data are within the manuscript and its Supporting information files.

## Abstract

### Introduction

Type 2 diabetes mellitus (T2DM), a chronic metabolic disorder, significantly increases cardiovascular disease (CVD) risk. Integrative care (IC) offers a personalized health management approach, utilizing various interventions to mitigate this risk. However, the impact of IC on CVD risk in newly diagnosed T2Dm patients remains unclear. This study aims to assess the differences in CVD risk development within 120 months following a new diagnosis of T2DM, using real-world data from Bumrungrad International Hospital and Vitallife Scientific Wellness Center.

### Methods

This study utilized the BI-VitalLife Cohort dataset that contains de-identified demographics, vitals, diagnoses and clinical information, laboratory and radiological data, medications, and treatments of more than 2.8 million patients who visited Bumrungrad International Hospital and/or VitalLife Scientific Wellness Center from June 1, 1999, to May 31, 2022. This study focused on newly diagnosed T2DM patients, defined according to American Diabetes Association criteria. We compared CVD risk between the IC and conventional care (CC) groups using the Kaplan-Meier curve and Cox proportional hazard model, adjusted for age, sex, and laboratory values. Propensity score matching was employed to enhance comparability.

### Results

Of the 5,687 patients included, 236 were in the IC group and 5,451 in the CC group. The IC group, characterized by a lower age at T2DM diagnosis, showed favorable hematological

**Funding:** The author(s) received no specific funding for this work.

**Competing interests:** The authors have declared that no competing interests exist.

and metabolic profiles. The Cox proportional hazard ratios revealed a significantly lower CVD risk in the IC group within 120 months post-T2DM diagnosis compared to the CC group, consistent even after adjusting for confounding factors. Propensity score-matched analysis supported these findings.

## Conclusion

Personalized integrative care may offer a significant advantage in reducing CVD risk among newly diagnosed T2DM patients compared to conventional care, even when considering various confounding factors. This study sheds light on the potential of integrative care in informing treatment strategies for T2DM patients at risk of developing CVD.

## Introduction

Individuals with type 2 diabetes mellitus (T2DM) are at increased risk of developing cardiovascular diseases (CVD), a leading global cause of death [1, 2]. Crucial to mitigating this risk are early detection and lifestyle changes, including diet adaptations [3, 4]. Conventional clinical care (CC) for T2DM, typically episodic and illness-focused, follows established guidelines for patient management [5, 6]. However, Integrative Care (IC) merges conventional and complementary medicine, offering a more holistic, person-centric approach [7]. IC emphasizes continuous care and lifestyle modifications such as diet, exercise, and other non-pharmacological interventions to reduce CVD risk [8–11].

The growing popularity of IC lies in its personalized approaches, adapting to individual patient needs. However, the diverse range of therapies and interventions within IC poses unique challenges in clinical study design and the measurement of outcomes, especially in retrospective studies [9–11]. Bumrungrad International Hospital (BIH) and VitalLife Scientific Wellness Center (VSWC) in Bangkok, Thailand, are at the forefront of offering an integrative approach to wellness care.

Utilizing the BI-VitalLife Cohort, a comprehensive dataset from these institutions, this study aims to explore the impact of IC in preventing major diseases among patients newly diagnosed with T2DM. Specifically, the primary objective of this study is to examine the differences in CVD risk within 120 months after T2DM diagnosis among those receiving IC compared to those undergoing episodic care alone. The choice of this 120-month duration for our study is inspired by the ASCVD 10-year prediction model [12], a recognized benchmark in cardiovascular research. This timeframe allows us to provide a long-term perspective on the effectiveness of IC in managing CVD risk among T2DM patients, aligning our research with established methodologies in the field. By focusing on this time period, we aim to offer insights into the sustained impacts of IC over a decade, crucial for understanding its long-term benefits in a real-world setting.

## Methods

This retrospective observational cohort study focused on newly diagnosed T2DM patients, utilizing the BI-VitalLife Cohort dataset (S1 Dataset). This dataset includes de-identified demographics, vitals, diagnoses, clinical information, laboratory and radiological data, medications, and treatments for over 2.8 million patients who visited Bumrungrad International Hospital and/or VitalLife Scientific Wellness Center from June 1, 1999, to May 31, 2022. The data were fully anonymized prior to access. The study has been exempted by the Bumrungrad International-Institutional Review Board (BI-IRB) (21-05-294FIEK-B) with no requirement for

written informed consent due to the retrospective nature of the study. Also, this study has been carefully conducted in compliance with the Thai Personal Data Protection Act (PDPA) that was announced on May 27, 2019, but has been postponed to enforcement on June 1, 2022.

## Determination of incident diabetes mellitus

T2DM was defined in our study as patients who had two consecutive glycemic criteria (GC) that met one of the four laboratory values in the corresponding diabetes ranges outlined by the American Diabetes Association (ADA) guidelines (Normal: HbA1c < 5.7% and Fasting Plasma Glucose (FPG) < = 99 mg/dL and Oral Glucose Tolerance Test (OGTT) < = 139; Pre-diabetes: HbA1c 5.7–6.4% or FPG 100–125 mg/dL or OGTT 140–199 mg/dL; Diabetes mellitus: HbA1c > = 6.5% or FPG > = 126 mg/dL or OGTT > = 200 mg/dL or Random Plasma Glucose > = 200 mg/dL) [13]. While the ADA recommends performing the second test the following day, this was not always available in out retrospective cohort dataset. Therefore, we considered all patients with at least two consecutive glycemic tests, one of which must be FPG or HbA1c, who were in the diabetic range within 90 days as eligible for our study. The date of diagnosis of T2DM is the date of the first GC event. Individuals who had received an anti-diabetic agent at the time of diagnosis were excluded as it influences the actual date of diagnosis. Individuals with laboratory values in the GC levels as the first values are also excluded from the studies since the date of onset is indeterminate.

## Determination of CVD onset

The date of onset of CVD was defined as the first recorded date of the International Classification of Diseases, Tent Revision (ICD-10) code, ranging from I00 to I99 [14]. To avoid potential confounding effects, we excluded patients who had received their first CVD diagnosis before their T2DM diagnosis.

## Operational definition of integrative care

As IC is an iterative and continuous engagement between the clinician and the client, one of the challenges with this approach is objectively measuring the client's compliance. However, continuous involvement in IC can make each integrative care visit more personalized and cumulatively effective. Various components of integrated care implemented at VSWC included but not limited to:

1. Collaborative Care Planning: Our team of healthcare professionals, including physicians, nurses, and specialists, work together to develop a personalized care plan for each patient. This collaborative approach ensures that all aspects of a patient's health, including but not limited to dietary habits, exercise habits, current lab values, and/or on-going supplements and medications, are taken into consideration, leading to a comprehensive and holistic care plan.

2. Care Coordination and Communication: We prioritize effective communication and coordination among different healthcare providers involved in a patient's care. This involves sharing information, exchanging insights, and actively involving the patient in decision-making processes. Regular team meetings and electronic health records facilitate seamless coordination and continuity of care.

3. Continuity of Care: We emphasize the importance of continuity in healthcare delivery. This involves maintaining a long-term relationship between patients and their healthcare

providers, ensuring consistent monitoring, follow-up, and adjustment of the care plan as needed.

4. Patient-Centric Approach: Our integrated care model places the patient at the center, recognizing their unique values, preferences, and goals. We engage patients in shared decision-making, empower them to actively participate in their own care, and provide education and support to promote self-management and well-being.

To assess compliance with wellness and disease management, we used the number of visits to VSWC and BIH as a proxy in our study, given the nature of the BI-VitalLife Cohort dataset. We excluded VSWC visits for aesthetic purposes, which were defined by the service provided and/or the medication prescribed or administered. For instance, a VSWC visit for botulinum toxin injection for wrinkle reduction was excluded. Individuals with zero visits to the VSWC were classified as the CC group, while others were classified as the IC group. To explore potential dose-response patterns, we divided the number of VSWC visits into quartiles.

## Adjustment variables

In our efforts to minimize potential confounding effects, the study strategically adjusted for key demographic variables, including age and sex. Additionally, we incorporated a range of significant laboratory variables to enhance the accuracy of our comparisons between the CC and IC in terms of their effectiveness on CVD outcomes. These laboratory variables included fasting blood glucose, HbA1c levels, and lipid profile parameters (LDL, HDL, and triglycerides), along with renal function indicators like serum creatinine and estimated glomerular filtration rate (eGFR). These variables were chosen due to their established clinical relevance in the progression of Type 2 Diabetes Mellitus (T2DM) and the associated cardiovascular risks. By adjusting for these specific variables, our analysis aims to provide a more precise and reliable evaluation of the impact of IC compared to CC in the context of CVD risk among T2DM patients.

## Statistical analysis

The study employed various statistical techniques to compare the risk of developing CVD between two groups. Descriptive statistics were used to present the characteristics of the individuals in the dataset, while the Kaplan-Meier curve was used to compare CVD risk between the two groups. Cox proportional hazard analysis was performed to compare the risk of developing CVD, adjusting for age, sex, and various laboratory values. Propensity score matching was also used to estimate the effect of one treatment (IC) compared to the other (CC), by taking into account the covariates that predict the likelihood of receiving IC or CC. The propensity score was calculated using logistic regression and a 1:4 or 1:2 nearest neighbor matching algorithm, meaning that each participant in the IC group was matched with four or two participants in the CC group, respectively. Only covariates with a sufficient sample size were included in the analysis.

## Results

### Characteristics of the participants

The study analyzed data from 5,687 patients (108,415 visits), with 236 patients in the intervention (IC) group and 5,451 in the control (CC) group (Table 1). The mean age at diagnosis of T2DM and CVD were 55.84±13.49 and 61.85±12.18 years old, respectively. The IC group had a significantly lower mean age at diagnosis of T2DM than the CC group (51.00±13.10 vs 56.05

**Table 1. Characteristics of T2DM Individuals in the BI-VitalLife Cohort.**

|  | Overall | Conventional Care | Integrative Care | p-value |
|---|---|---|---|---|
| # Individuals | 5,687 | 5,451* | 236 | |
| • Inpatient | | 1,175 | 0 | |
| • Outpatient | | 5,450 | 236 | |
| # Visits | 108,415 | 107,190 | 1,225 | |
| • Inpatient | 2,737 | 2,737 | 0 | |
| • Outpatient | 105,678 | 104,453 | 1,225 | |
| Age at T2DM Diagnosis (years; Mean±SD) | 55.84±13.49 | 56.05±13.47 | 51.00±13.10 | <0.001 |
| Female (%) | 39.51 | 39.42 | 41.52 | 0.518 |
| CVD cases (n) | 1,945 | 1,901 | 44 | |
| Age at CVD Diagnosis | 61.85±12.18 | 62.87±12.14 | 61.11±13.97 | 0.685 |
| (years; Mean±SD) | | | | |
| CVD in 120 months (%) | 32.81 | 33.57 | 15.25 | <0.001 |

* One patient received both inpatient and outpatient care.

±13.47, p<0.001), higher hematocrit (41.73±4.93 vs 40.72±5.45, p = 0.008) and hemoglobin levels (13.94±1.74 vs 13.58±1.95, p = 0.009), lower white blood cells (7.17±2.59 vs 7.94±4.32, p = 0.007) and neutrophil counts (4,240±230 vs 4,970±360, p = 0.002), higher total cholesterol (204.14±55.91 vs 196.39±52.78, p = 0.034), better kidney function (creatinine 0.86±0.90 vs 0.98 ±0.74, p = 0.019; estimated glomerular filtration rate [eGFR] 100.68±21.69 vs 90.82±25.35, p<0.001) and lower erythrocyte sedimentation rate (ESR 20.39±18.69 vs 35.64±30.55, p<0.001) (Table 2).

## Kaplan-Meier survival analysis: IC vs. CC

The risk of developing CVD within 120 months after being diagnosed with T2DM was significantly lower in the IC group than in the CC group, as demonstrated by Kaplan-Meier survival analysis ($p = 3.89 \times 10^{-6}$) (Fig 1). Stratification by sex shows statistical significance for both women and men ($p = 2.58 \times 10^{-3}$ and $4.51 \times 10^{-4}$, respectively) (Fig 2). When stratified by age at diagnosis, the risk of developing CVD was significantly lower in the IC group than in the CC group for patients diagnosed with T2DM between the ages of 40 and 60 ($p = 3.32 \times 10^{-5}$), but not for those diagnosed after the age of 60 ($p = 3.10 \times 10^{-1}$) (Fig 3).

## Cox proportional hazard: IC vs. CC

In assessing the effectiveness of IC compared to CC, our analysis utilized Cox proportional hazard models, as detailed in Table 3. The hazard ratios (HRs) revealed a statistically significant protective effect of IC against the development of CVD in T2DM patients. Specifically, Model 0, which did not include any adjustments, showed an HR of 0.460, indicating a strong protective effect of IC. This effect persisted across Model 1 to 6, which incorporated adjustments for age at T2DM diagnosis, sex, creatinine, hemoglobin, neutrophils, and cholesterol, with HRs ranging from 0.502 to 0.516. This slight increase in HRs in the adjusted models suggests a moderate dilution effect attributable to the confounding variables.

Furthermore, age was identified as a significant factor; the risk of developing CVD increased with age, with HRs ranging from 0.022 to 0.025 for each additional year (p<0.001 across Models 1–6). Additionally, creatinine levels also inluenced CVD risk, with an increase of approximately 0.10 to 0.15 in HR for each additional mg/dl of creatinine (p<0.001 in

**Table 2. Laboratory characteristics at the diagnosis of T2DM in the BI-VitalLife Cohort.**

| | Overall | Conventional Care | n | Integrative Care | n | p-value |
|---|---|---|---|---|---|---|
| Hematocrit (%) | 40.76 ± 5.44 | 40.72 ± 5.45 | 4,924 | 41.73 ± 4.93 | 207 | 0.008 |
| Hemoglobin (g/dl) | 13.60 ± 1.94 | 13.58 ± 1.95 | 4,939 | 13.94 ± 1.74 | 212 | 0.009 |
| Red Blood Count (RBC) (x10$^6$/ul) | 4.76 ± 0.74 | 4.75 ± 0.74 | 5,113 | 4.97 ± 0.69 | 230 | <0.001 |
| Mean Corpuscular Hemoglobin (MCH) (pg) | 28.79 ± 3.48 | 28.80 ± 3.50 | 5,113 | 28.51 ± 3.05 | 230 | 0.213 |
| Mean Corpuscular Volume (MCV) (fl) | 82.90 ± 7.23 | 82.96 ± 7.26 | 812 | 82.18 ± 6.97 | 76 | 0.371 |
| Mean Corpuscular Hemoglobin Concentration (MCHC) (g/dl) | 33.44 ± 1.15 | 33.43 ± 1.15 | 5,111 | 33.63 ± 1.19 | 230 | 0.012 |
| Red Cell Distribuion idth (RDW)(%) | 14.00 ± 2.17 | 14.02 ± 2.19 | 5,113 | 13.67 ± 1.82 | 230 | 0.018 |
| White Blood Count (WBC) (x10$^3$/ul) | 7.91 ± 4.27 | 7.94 ± 4.32 | 5,113 | 7.17 ± 2.59 | 230 | 0.007 |
| Neutrophil (%) | 59.51 ± 13.62 | 59.60 ± 13.71 | 5,112 | 57.45 ± 11.50 | 230 | 0.019 |
| Neutrophil (x10$^3$/ul) | 4.94 ± 3.56 | 4.97 ± 3.60 | 5,112 | 4.24 ± 2.30 | 230 | 0.002 |
| Lymphocyte (%) | 29.69 ± 11.87 | 29.60 ± 11.93 | 5,112 | 31.64 ± 10.23 | 230 | 0.011 |
| Lymphocyte (x10$^3$/ul) | 2.13 ± 0.98 | 2.13 ± 0.99 | 5,112 | 2.16 ± 0.83 | 230 | 0.636 |
| Eosinophil (%) | 3.05 ± 2.45 | 3.04 ± 2.45 | 5,037 | 3.24 ± 2.39 | 229 | 0.226 |
| Eosinophil (x10$^3$/ul) | 0.23 ± 0.23 | 0.23 ± 0.24 | 5,037 | 0.23 ± 0.19 | 229 | 0.917 |
| Basophil (%) | 0.68 ± 0.40 | 0.68 ± 0.40 | 4,921 | 0.70 ± 0.37 | 229 | 0.420 |
| Platelet (x10$^3$/ul) | 253.01 ± 86.33 | 252.78 ± 87.01 | 5,113 | 258.14 ± 69.41 | 230 | 0.356 |
| Creatinine (mg/dl) | 0.98 ± 0.74 | 0.98 ± 0.74 | 5,272 | 0.86 ± 0.90 | 232 | 0.019 |
| Estimated glomerular filtration rate (eGFR) (ml/min/1.73m$^2$) | 91.24 ± 25.28 | 90.82 ± 25.35 | 5,272 | 100.68 ± 21.69 | 232 | <0.001 |
| Aspartate Aminotransferase (AST) (u/l) | 34.43 ± 57.37 | 34.6 ± 58.36 | 4,823 | 30.62 ± 27.19 | 218 | 0.317 |
| Alanine Transaminase (ALT) (u/l) | 44.81 ± 137.94 | 44.93 ± 140.86 | 4,943 | 42.14 ± 35.88 | 227 | 0.766 |
| Gamma-glutamyl Transferase (GGT) (u/l) | 73.28 ± 135.39 | 73.7 ± 136.93 | 3,583 | 64.28 ± 97.02 | 169 | 0.377 |
| C-Reactive Protein (CRP)(mg/dl) | 8.95 ± 37.57 | 10.01 ± 40.23 | 389 | 2.06 ± 4.32 | 60 | 0.127 |
| Erythrocyte Sedimentation Rate (ESR) (mm/h) | 34.24 ± 29.97 | 35.64 ± 30.55 | 582 | 20.39 ± 18.69 | 59 | <0.001 |
| Homocysteine (umol/l) | 10.50 ± 4.36 | 11.09 ± 4.86 | 52 | 8.98 ± 2.03 | 20 | 0.066 |
| Sodium (mmol/l) | 137.59 ± 4.36 | 137.58 ± 4.37 | 2,390 | 137.90 ± 4.11 | 104 | 0.463 |
| Potassium (mmol/l) | 4.19 ± 0.52 | 4.19 ± 0.53 | 2,551 | 4.17 ± 0.41 | 109 | 0.680 |
| Cholesterol (mg/dl) | 196.73 ± 52.94 | 196.39 ± 52.78 | 4,684 | 204.14 ± 55.91 | 219 | 0.034 |
| Low-Density Lipoprotein (LDL) cholesterol (mg/dl) | 120.31 ± 40.36 | 120.07 ± 40.4 | 4,397 | 125.09 ± 39.35 | 213 | 0.077 |
| Triglycerides (mg/dl) | 180.09 ± 267.56 | 180.46 ± 272.02 | 4,686 | 172.04 ± 140.94 | 218 | 0.650 |
| High-Density Lipoprotein (HDL) cholesterol (mg/dl) | 45.36 ± 13.91 | 45.32 ± 14.00 | 4,523 | 46.14 ± 11.93 | 217 | 0.398 |
| Fasting Blood Glucose (FBG) (mg/dl) | 163.42 ± 58.55 | 163.55 ± 58.64 | 5,196 | 160.26 ± 56.57 | 227 | 0.407 |

Models 4–6). Notably, in the adjusted models, the different in total cholesterol levels between the IC and CC groups was not statistically significant.

## Cox proportional hazard: Quartiles of VSWC visits

Our study also explored the 'dose response' relationship between the frequency of IC visits to VSWC and CVD protection in T2DM patients. This aspect was analyzed by categorizing IC visits into quartiles and examining their impact on CVD risk using Cox proportional hazard models, as detailed in Table 4.

The analysis revealed a notable trend: with each increase in the quartile of VSWC visits, indicative of more frequent IC engagement, there was a significantly reduction in CVD risk. Specifically, in Model 0 (unadjusted), for each higher quartile of visits, the HR for developing CVD decreased by 0.175 (HR 0.825, p = 0.003), suggesting a substantial protective effect of more frequent IC visits.

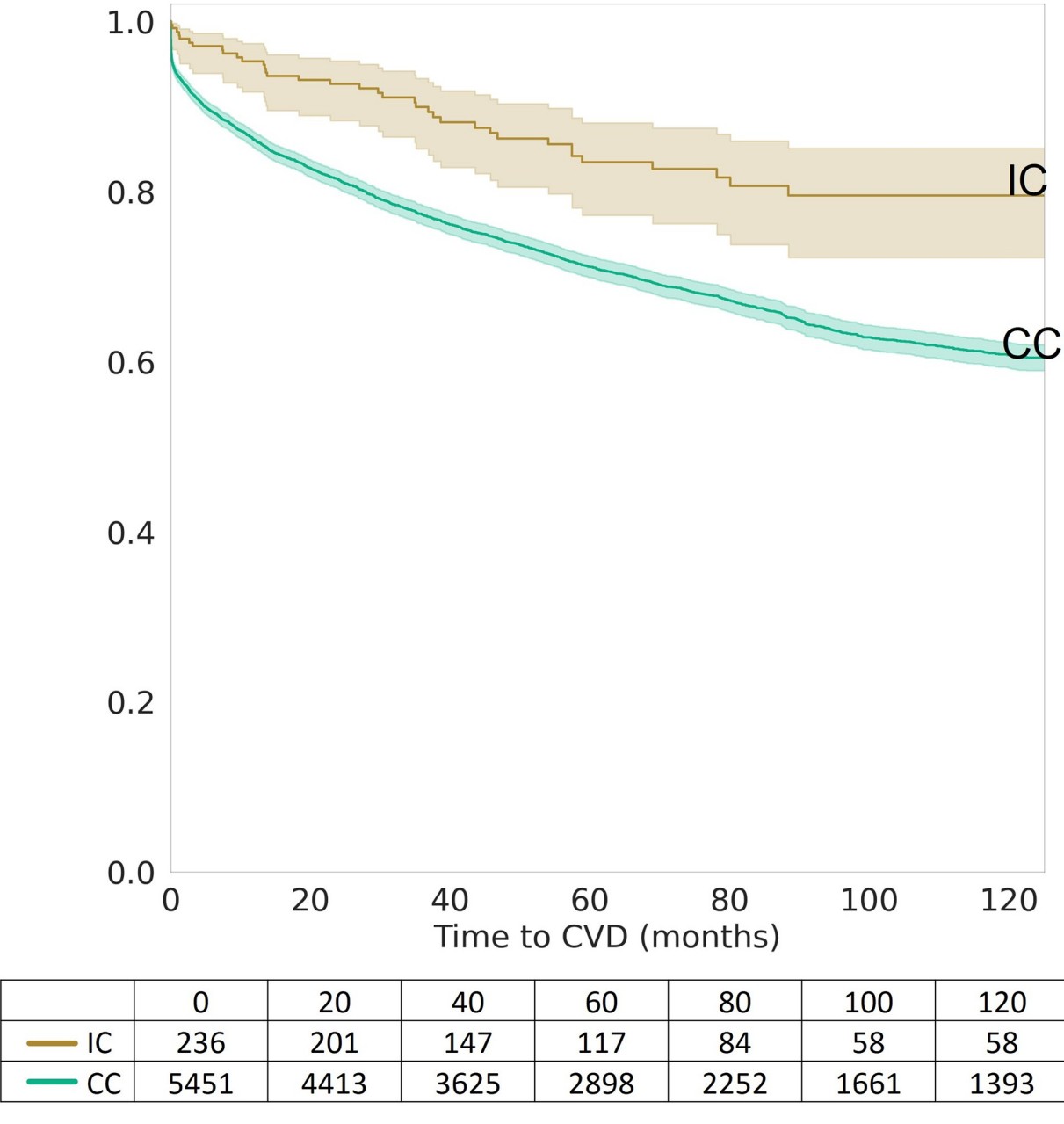

| | | 0 | 20 | 40 | 60 | 80 | 100 | 120 |
|---|---|---|---|---|---|---|---|---|
| —— | IC | 236 | 201 | 147 | 117 | 84 | 58 | 58 |
| —— | CC | 5451 | 4413 | 3625 | 2898 | 2252 | 1661 | 1393 |

**Fig 1. Kaplan-Meier survival curves for integrative care (IC) vs. conventional care (CC).** Kaplan-Meier survival curves depicting the difference in cardiovascular disease (CVD) risk over 120 months post-diagnosis of type 2 diabetes mellitus (T2DM). The curves compare the survival probabilities of patients receiving Integrative Care (IC) versus those receiving Conventional Care (CC), with the number of patients at risk provided at various time intervals. The IC group demonstrates improved survival rates over time compared to the CC group.

This protective relationship persisted even after adjusting for confounding variables such as age, sex, hemoglobin, neutrophils, and cholesterol in Models 1–6. In these adjusted models, the HRs ranged between 0.854 and 0.859 (p-values raning from 0.016 to 0.020, indicating that the increased frequency of IC visits continued to be significantly associated with a lower risk of developing CVD.

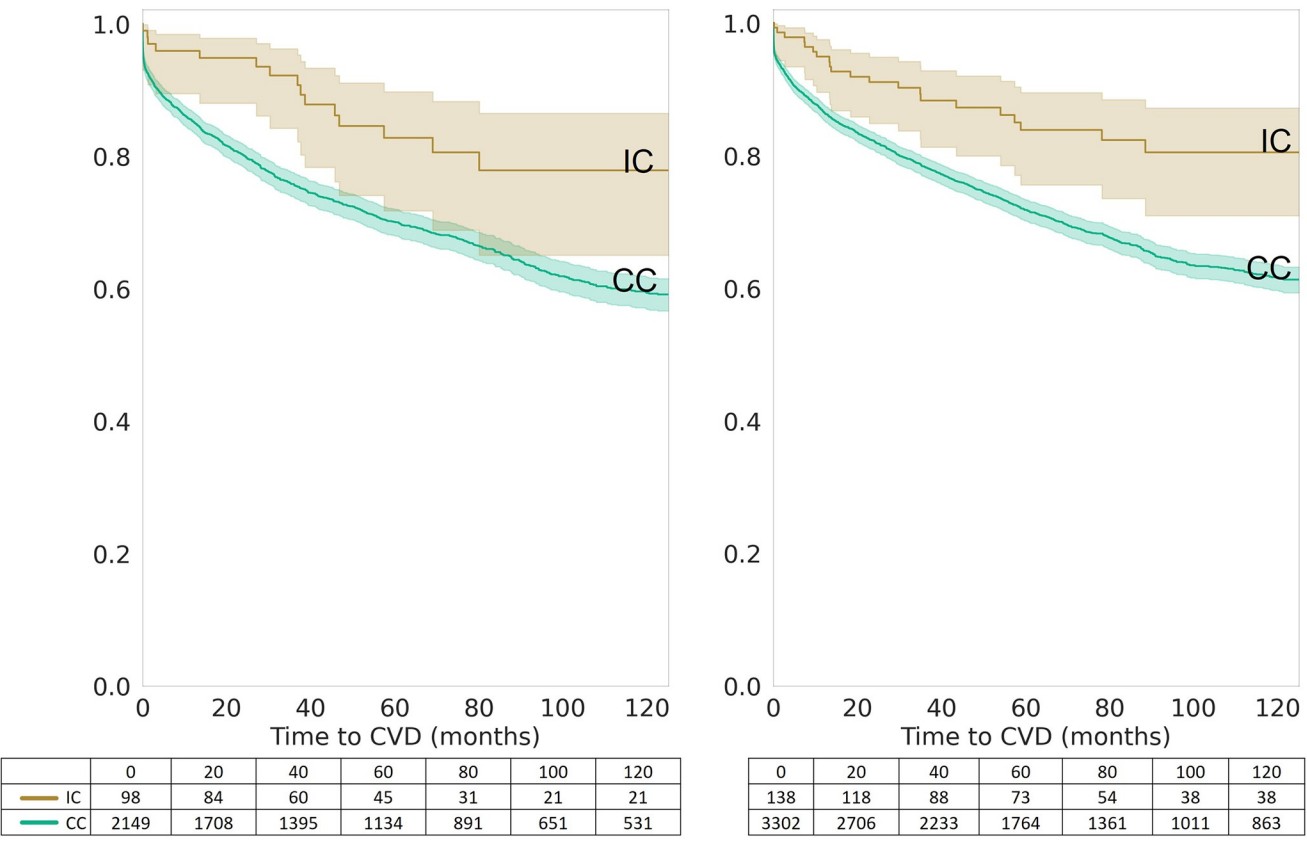

**Fig 2. Sex-specific Kaplan-Meier survival curves for integrative care (IC) vs. conventional care (CC).** Kaplan-Meier survival curves stratified by sex, comparing the survival probabilities of male and female patients receiving Integrative Care (IC) versus those receiving Conventional Care (CC) over a 120-month period post-Type 2 Diabetes Mellitus (T2DM) diagnosis. The curves indicate a consistently higher survival probability for patients in the IC group across both sexes, with the number of patients at risk displayed at standard time intervals, reflecting the sex-specific influence on the effectiveness of IC in reducing CVD risk.

## Cox proportional hazard with propensity score matching

To facilitate a more accurate comparison between IC and CC, our analysis employed a fully adjusted model (Model 6) along with propensity score matching. This approach utilized both 1:4 and 1:2 nearest neighbor matching ratios, aiming to balance the distribution of clinical and demographic variables between the two groups. The details of these comparisons, including the variables' balances before and after propensity score matching, are presented in S1 Table.

Initially, significant disparities were noted in variables such as age at T2DM diagnosis, RBC, WBC, Neutrophil count and percentage, Lymphocyte percentage, estimated glomerular filtration rate (eGFR), and mean corpuscular hemoglobin concentration (MCHC). However, post-propensity score matching, an increase in p-values was observed, indicating a more balanced distribution between IC and CC groups. Interestingly, matching ratios of 1:4 and 1:2 yielded similar results, suggesting that the number of matched controls did not markedly influence the outcomes.

Propensity score matching effectively aligned the groups on observed covariates, thus allowing for a more robust comparison of IC and CC effects. The consistent Cox proportional hazard ratio of 0.512, even after adjustments, underscores the protective effect of IC relative to CC, thereby reinforcing the study's validity.

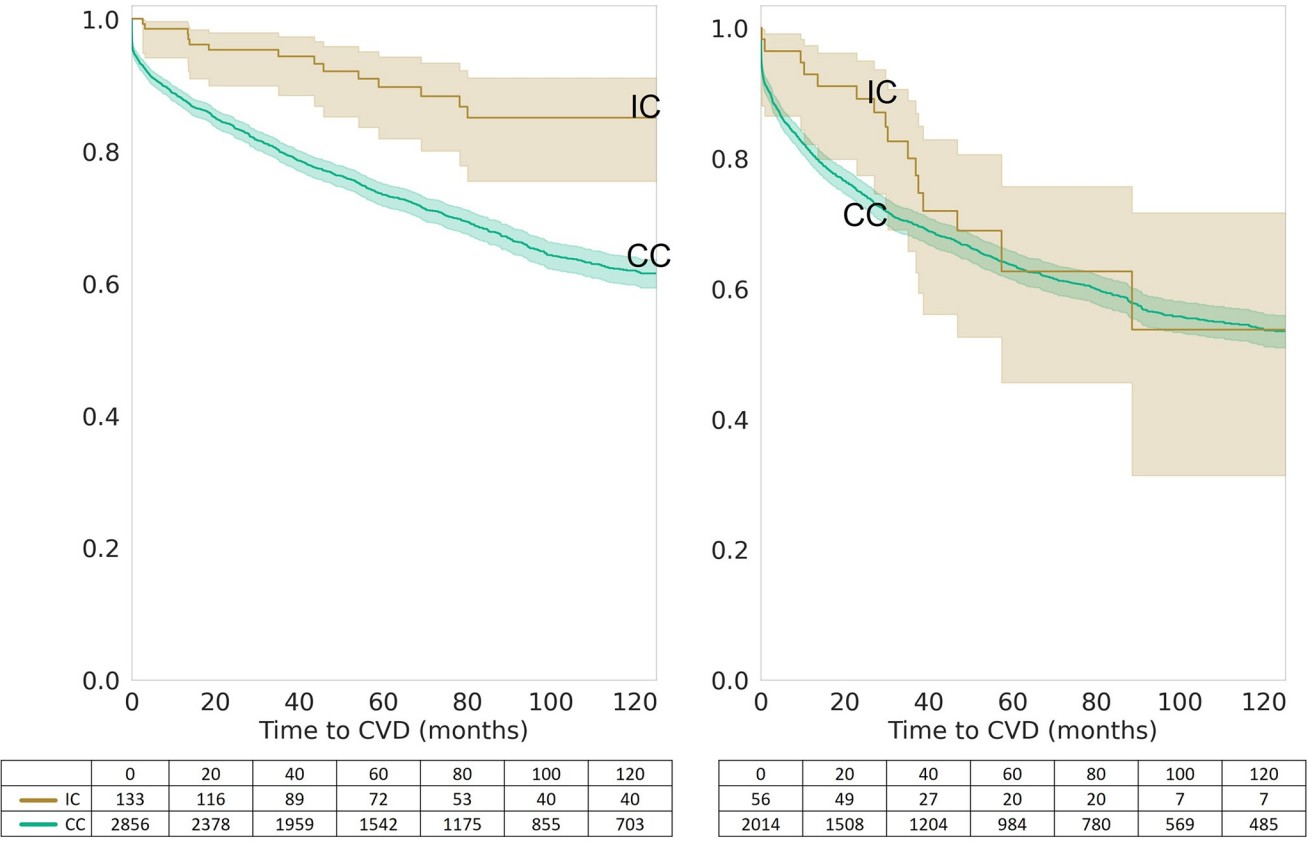

**Fig 3. Age-stratified Kaplan-Meier survival curves for integrative care (IC) vs. conventional care (CC).** Kaplan-Meier survival curves stratified by age, illustrating the survival probabilities of patients receiving Integrative Care (IC) versus those receiving Conventional Care (CC) over a 120-month period post-Type 2 Diabetes Mellitus (T2DM) diagnosis. These curves display the survival outcomes for different age groups within the study population, with a table below indicating the number of patients at risk at specified intervals. The comparison highlights the nuanced impact of IC on CVD risk reduction across various age demographics.

**Table 3. Cox proportional hazard analysis comparing integrative care vs conventional care.**

| | N Total | N CC | N IC | VTL | Age at Diagnosis of T2DM | Male | Creatinine | Hemoglobin | Neutrophil | Cholesterol |
|---|---|---|---|---|---|---|---|---|---|---|
| Model 0 | 5687 | 5451 | 236 | 0.46 (<0.001; [0.331, 0.639]) | - | - | - | - | - | - |
| Model 1 | 5687 | 5451 | 236 | 0.516 (<0.001; [0.371, 0.718]) | 1.023 (<0.001; [1.019, 1.026]) | - | - | - | - | - |
| Model 2 | 5687 | 5451 | 236 | 0.516 (<0.001; [0.37, 0.717]) | 1.023 (<0.001; [1.019, 1.026]) | 0.969 (0.503; [0.883, 1.063]) | - | - | - | - |
| Model 3 | 5504 | 5272 | 232 | 0.515 (<0.001; [0.369, 0.721]) | 1.022 (<0.001; [1.019, 1.026]) | 0.953 (0.33; [0.865, 1.05]) | 1.055 (0.055; [0.999, 1.115]) | - | - | - |
| Model 4 | 5095 | 4883 | 212 | 0.51 (<0.001; [0.361, 0.72]) | 1.024 (<0.001; [1.02, 1.028]) | 0.858 (0.008; [0.767, 0.96]) | 1.102 (0.001; [1.038, 1.17]) | 1.066 (<0.001; [1.036, 1.097]) | - | - |
| Model 5 | 5094 | 4882 | 212 | 0.502 (<0.001; [0.356, 0.71]) | 1.024 (<0.001; [1.02, 1.028]) | 0.86 (0.009; [0.768, 0.962]) | 1.116 (<0.001; [1.052, 1.185]) | 1.059 (<0.001; [1.028, 1.09]) | 0.976 (0.002; [0.962, 0.991]) | - |
| Model 6 | 4452 | 4255 | 197 | 0.505 (<0.001; [0.356, 0.718]) | 1.025 (<0.001; [1.02, 1.029]) | 0.851 (0.01; [0.753, 0.962]) | 1.155 (<0.001; [1.084, 1.23]) | 1.053 (0.002; [1.019, 1.09]) | 0.988 (0.195; [0.97, 1.006]) | 1.0 (0.521; [0.999, 1.001]) |

**Table 4. Cox proportional hazard analysis comparing the quartiles of VSWC visits.**

| | N | N | N | VSWSC Quartiles | Age at Diagnosis of T2DM | Male | Creatinine | Hemoglobin | Neutrophil | Cholesterol |
|---|---|---|---|---|---|---|---|---|---|---|
| | Total | CC | IC | | | | | | | |
| Model 0 | 5687 | 5451 | 236 | 0.825 (0.003; [0.727, 0.935]) | - | - | - | - | - | - |
| Model 1 | 5687 | 5451 | 236 | 0.859 (0.016; [0.759, 0.972]) | 1.023 (<0.001; [1.02, 1.027]) | - | - | - | - | - |
| Model 2 | 5687 | 5451 | 236 | 0.859 (0.016; [0.759, 0.972]) | 1.023 (<0.001; [1.02, 1.027]) | 0.969 (0.507; [0.883, 1.063]) | - | - | - | - |
| Model 3 | 5504 | 5272 | 232 | 0.856 (0.016; [0.754, 0.971]) | 1.023 (<0.001; [1.019, 1.026]) | 0.953 (0.331; [0.866, 1.05]) | 1.057 (0.049; [1.0, 1.116]) | - | - | - |
| Model 4 | 5095 | 4883 | 212 | 0.859 (0.02; [0.755, 0.976]) | 1.024 (<0.001; [1.02, 1.028]) | 0.86 (0.008; [0.768, 0.962]) | 1.103 (0.001; [1.039, 1.171]) | 1.065 (<0.001; [1.035, 1.097]) | - | - |
| Model 5 | 5094 | 4882 | 212 | 0.854 (0.016; [0.751, 0.971]) | 1.024 (<0.001; [1.021, 1.028]) | 0.861 (0.009; [0.769, 0.964]) | 1.116 (<0.001; [1.052, 1.185]) | 1.058 (<0.001; [1.027, 1.09]) | 0.977 (0.002; [0.962, 0.992]) | - |
| Model 6 | 4452 | 4255 | 197 | 0.856 (0.018; [0.752, 0.974]) | 1.025 (<0.001; [1.021, 1.029]) | 0.852 (0.01; [0.753, 0.963]) | 1.154 (<0.001; [1.084, 1.229]) | 1.053 (0.003; [1.018, 1.089]) | 0.989 (0.215; [0.971, 1.007]) | 1.0 (0.519; [0.999, 1.001]) |

Furthermore, while no significant differences in age at CVD diagnosis were initially found between IC and CC groups, propensity score matching based on age at T2DM diagnosis painted a different picture. It revealed that IC conferred a greater protective effect than CC, emphasizing the critical role of age as a confounding factor in our analysis.

## Discussion

This 20-year retrospective analysis provides compelling evidence that IC significantly reduce the risk of CVD development in patients with T2DM. Patients who engaged in IC at both BIH and VSWC demonstrated a markedly lower incidence of CVD compared to those received only CC at BIH. This key finding highlights the effectiveness of IC's comprehensive, patient-centered approach over the more segmented and disease-specific focus of traditional CC. By integrating lifestyle management, including diet and exercise modifications, as a core component of therapy, IC offers a proactive strategy to mitigate one of the primary health threats faced by individuals with T2DM.

The remainder of the discussion explores the broader context and implications of this finding. While conventional specialists often focus narrowly on their specific roles in patient care, the IC model prioritizes a holistic view of the patient's health, aligning with findings from landmark studies like the Framingham Heart Study and EGAT regarding the root causes and predictive risks of CVD [15, 16]. This approach underlines the importance of patient participation and lifestyle management in preventing CVD among T2DM patients [17]. Additionally, our study suggests that the success of IC does not require an extensive array of specialized professionals; rather, a small, well-coordinated team can effectively implement personalized care plans. This finding supports the viability of IC in various healthcare settings and underscores its potential in improving long-term health outcomes for T2DM patients.

Notably, the success of IC does not depend on the involvement of specialized professionals alone. A recent meta-analysis revealed that integrated interventions, including an active endocrinologist, led to moderate improvements in HbA1c, blood pressure, and weight management; however, specific CVD outcomes were not addressed [18]. Moreover, multicomponent integrated care based on the Chronic Care Model (CCM) demonstrated modest improvements in several surrogate parameters, such as HbA1c, blood pressure, and lipid profiles, according

to a meta-analysis published in 2018 [19]. In the context of our study, IC does not necessitate a large team or an array of professional disciplines. Instead, a small team led by a wellness physician can effectively implement IC by tailoring multiple small interventions to meet the unique needs and preferences of each individual, leading to a course of personalized care with a focus on achieving holistic outcomes. By adopting this approach, we anticipate cumulative and meaningful improvements in the overall health of patients. The findings from our study emphasize that IC can be implemented by a well-organized team, combining the expertise of a wellness physician and other healthcare professionals, without the requirement for an extensive specialized setup. This approach ensures individualized care that addresses specific health concerns, ultimately contributing to better long-term health outcomes for patients with T2DM. Nevertheless, further research and clinical trials are warranted to examine specific CVD outcomes in the context of IC to provide a more comprehensive understanding of its impact on cardiovascular health.

While the benefits of personalized continuum and integrative care may vary between wellness centers or personalized regimens, additional evidence from randomized controlled trials (RCTs) is necessary to establish their effectiveness. Despite this, the present study provides significant insights into the adjunctive benefits of IC for patients with chronic conditions. However, designing an RCT with consistent interventions and controls, participant compliance, and measurable outcomes may limit the external generalizability of the results to other settings. In addition to lifestyle management, the VSWC program incorporates preventive approaches based on eight of the nine 'Hallmarks of Aging', including telomere attrition, epigenetic alterations, loss of proteostasis, deregulated nutrient sensing, mitochondrial dysfunction, cellular senescence, stem cell exhaustion, and altered intercellular communication [18]. The selection of specific interventions that correspond to CVD risk reduction among patients with T2DM who have undergone integrative BIH-VSWC care provides promising directions for future validation studies. To further improve our analysis, further studies will incorporate the remaining hallmark of aging, genomic instability [19, 20]. These findings will contribute to a better understanding of the potential benefits of personalized and integrative care for patients with chronic conditions.

The success of personalized continuum and integrative care relies on regular patient follow-up, engagement, and compliance. While the longitudinal data from BI-VitalLife is extensive, the dataset lacks objective measurements of patient participation and adherence to the recommended regime, which limits the ability to derive definitive conclusions. Currently, there are no strict policies or guidelines to determine whether a patient should be included in IC, as all IC clients are voluntarily engaged. This may lead to a population bias, where individuals who participate in IC are already more health-conscious than participating in conventional care. Future studies could benefit from incorporating more rigorous methods of assessing patient participation and adherence to the recommended regime. This could include the use of objective measures, such as wearable technology, to monitor patient behavior and engagement. Additionally, efforts could be made to recruit a more diverse patient population, including those who may not typically seek out integrative care services. By addressing these limitations, we can gain a better understanding of the potential benefits and limitations of personalized and integrative care for a broader patient population.

In this study, we used propensity score matching with a 1:4 ratio to match individuals in both groups based on a set of observed covariates, including age, gender, BMI, smoking status, and comorbidities. The choice of a 1:4 ratio was likely appropriate given the sample size and number of covariates. The sample size of 264 individuals was relatively small, and there were several covariates that needed to be accounted for. The 1:4 ratio allowed for a more robust matching process and improved balance between the two groups. However, it is important to

note that the use of PSM alone does not completely eliminate the potential for bias. There may still be unobserved covariates that differ between the two groups and can impact the results. Additionally, the use of PSM can potentially lead to a reduction in sample size, as some individuals may not be included in the analysis if they cannot be matched to an individual in the other group. In this study, the authors reported that 82 individuals were excluded from the analysis due to the inability to match them with an individual in the other group. This reduction in sample size may limit the generalizability of the results to a broader population. Overall, the use of propensity score matching with a 1:4 ratio was an appropriate method for this specific study, given the sample size and number of covariates. However, it is important to consider the potential limitations of PSM, including the possibility of unobserved covariates and reduced sample size, when interpreting the results.

The exclusion of pure aesthetic procedures from the definition of wellness care might be a controversial topic. While some aesthetic procedures may not be considered essential to overall health and well-being, they can offer psychological benefits by improving a person's appearance and self-esteem. However, these procedures may also carry certain risks and complications that could negatively affect a person's health and well-being. Wellness care, on the other hand, focuses on promoting and maintaining health and well-being in a holistic way, rather than solely addressing cosmetic concerns. It encompasses a range of practices, such as preventive care, lifestyle management, and disease management, that are designed to promote physical and mental well-being. It is important to note that there may be some overlap between aesthetic procedures and wellness care, as some treatment may offer therapeutic benefits in addition to cosmetic benefits. For example, certain facial treatments may improve skin health and reduce inflammation, leading to a healthier overall appearance. Overall, while the exclusion of pure aesthetic procedures from the definition of wellness care may be appropriate given the focus on promoting overall health and well-being, it is important to consider the potential benefits and risks of these procedures. There may be cases where an aesthetic procedure can contribute to a person's overall well-being and should be considered as part of their wellness care plan. Ultimately, the promotion of physical and mental health and well-being should remain the primary focus of wellness care, but it should also consider individual needs and preferences to provide comprehensive and personalized care.

## Conclusion

A study of 5,687 patients over 20 years found that personalized integrative care can lower the risk of developing cardiovascular disease (CVD) within 120 months of a type 2 diabetes diagnosis. Although the results were significant, further research is needed to validate the effectiveness of specific interventions. This study used big data and statistical analysis to provide new insights into the connection between integrative care and CVD, and it could help establish best practices for integrative medicine. Healthcare providers can use this information to make better treatment decisions for patients with T2DM and CVD. Additionally, these techniques could be adapted to identify recommendations for other diseases.

## Supporting information

**S1 Dataset. Bumrungrad-VitalLife Cohort dataset.**
(XLSX)

**S1 Table. Characteristics of the participants before and after 1:4 and 1:2 propensity score matching.**
(DOCX)

## Acknowledgments

We are grateful to Prof. Dr. Nimit Taechakraichana, Assoc. Prof. Dr. Suwatchai Pornratanar-angsi, and Dr. Palita Lungchukiet for their valuable guidance and support throughout the process.

## Author Contributions

**Conceptualization:** Tanawat Khunlertkit, Teeradache Viangteeravat, Jeremy Mark Ford.

**Data curation:** Tanawat Khunlertkit, Panupong Wangprapa.

**Formal analysis:** Tanawat Khunlertkit, Teeradache Viangteeravat, Panupong Wangprapa.

**Investigation:** Tanawat Khunlertkit, Teeradache Viangteeravat, Panupong Wangprapa, Krit Pongpirul.

**Methodology:** Tanawat Khunlertkit, Teeradache Viangteeravat, Krit Pongpirul.

**Project administration:** Teeradache Viangteeravat, Suthee Siriwechdaruk, Jeremy Mark Ford.

**Supervision:** Suthee Siriwechdaruk, Jeremy Mark Ford, Krit Pongpirul.

**Visualization:** Tanawat Khunlertkit, Panupong Wangprapa.

**Writing – original draft:** Tanawat Khunlertkit, Krit Pongpirul.

**Writing – review & editing:** Tanawat Khunlertkit, Teeradache Viangteeravat, Panupong Wangprapa, Suthee Siriwechdaruk, Jeremy Mark Ford, Krit Pongpirul.

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
