## [Decision Letter · Decision Letter 0]

6 Dec 2023

PONE-D-23-25653Potential Contribution of Wellness Care to Cardiovascular Disease Prevention among Diabetes Patients: Analysis of the BI-VitalLife Cohort.PLOS ONE

Dear Dr. Pongpirul,

Thank you for submitting your manuscript to PLOS ONE. After careful consideration, we feel that it has merit but does not fully meet PLOS ONE’s publication criteria as it currently stands. Therefore, we invite you to submit a revised version of the manuscript that addresses the points raised during the review process.

We look forward to receiving your revised manuscript.

Kind regards,

Amirmohammad Khalaji

Academic Editor

PLOS ONE

Journal Requirements:

2. Please include a separate caption for each figure in your manuscript.

Reviewers' comments:

Reviewer's Responses to Questions

**Comments to the Author**

1. Is the manuscript technically sound, and do the data support the conclusions?

Reviewer #1: Yes

Reviewer #2: Partly

2. Has the statistical analysis been performed appropriately and rigorously? 

Reviewer #1: Yes

Reviewer #2: Yes

3. Have the authors made all data underlying the findings in their manuscript fully available?

Reviewer #1: Yes

Reviewer #2: No

4. Is the manuscript presented in an intelligible fashion and written in standard English?

Reviewer #1: Yes

Reviewer #2: Yes

5. Review Comments to the Author

Reviewer #1: Khunlertkit et al. have performed a study on risk assessment of developing CVD after T2DM diagnosis. They found that personalized integrative care is associated with better care, compared to conventional care. Although it provides useful information to the field, there are some points that need to be mentioned.

MAJOR Remarks:

- In the abstract, the aim of the study should be stated at the end of the introduction/background.

- The introduction section is rather long. Authors should focus on the main ideas related to the manuscript.

- Figure captions are missed and should be added.

- The first paragraph of the discussion should focus on the most eye-catching findings of the current study. Please summarize the most important findings there.

MINOR Remarks:

- It is better to add the survival table and number of patients at risk during different time intervals below the Kaplan-Meier survival plots.

- In Tables 3 and 4, it is highly suggested to add 95% CI to the HRs provided.

Reviewer #2: The study is well-written and centers around an intriguing subject. However, certain aspects of the methodology require further clarification.

1- The concept of integrative care (IC) lacks clarity, as the authors do not provide specific details regarding the precise components of IC intervention in their methodology. The authors neglected to provide a comprehensive account of the specific measures taken for the patient population and their corresponding utilization rate. The study primarily focuses on various aspects of IC quality, such as patient compliance.

2- What was the author's rationale for considering the number of visits to VSWC as a proxy for compliance?

3- What is the author's rationale for excluding patients who had received an anti-diabetic agent, considering that they likely constitute a significant proportion of the diabetic population in any given community?

4- The relationship between cardiovascular disease (CVD) and diabetes mellitus (DM) is independent of which condition occurs first. However, it is certain that newly diagnosed DM can have a negative impact on the development and progression of previously diagnosed CVD. Considering this fact, the decision made by the authors to exclude patients who had been diagnosed with CVD prior to their type 2 diabetes mellitus (T2DM) diagnosis is debatable and requires clarification.

5- If the objective of the study is to assess newly diagnosed patients, this should be clearly stated in the title, abstract, and methodology.

6- The authors fail to specify the adjusted variables employed and merely mention the imprecise term "significant laboratory variables." The request is for a detailed presentation of the "significant laboratory variables" and an explanation of why they were selected as confounding variables.

7- Using “increased and decreased amounts” and ranges to represent the results cause confusion for readers.

8- In the section titled "Operational Definition of Integrative Care," the authors provide a list of IC goals as "Various components of integrated care," but they do not clearly define the specific intervention that was carried out.

9- Please provide a more explicit explanation of how the choice of a study duration of 120 months is connected to the inspiration derived from the ASCVD 10-year prediction and what is the significance of this particular time frame?

10- According to the ADA guidelines, it is recommended to perform the second glycemic criteria (GC) test on a separate day. Were the patients with two disrupted GCs on the same day considered diabetic?

11- The study objective is somewhat buried in the text. It is recommended to clearly state the main objective of the study in a distinct paragraph, and provide a concise summary of the study's design within it.

12- The use of terms such as "chronic metabolic 'silent killer'" in the introduction section might be considered sensationalistic and could be replaced with a more straightforward description of T2DM.

6. PLOS authors have the option to publish the peer review history of their article (what does this mean?). If published, this will include your full peer review and any attached files.

Reviewer #1: No

Reviewer #2: No

---

## [Author Response · Author response to Decision Letter 0]

17 Jan 2024

We thank you and the reviewers for the comments and suggestions. Please find our point-by-point responses below:

Reviewer 1: Khunlertkit et al. have performed a study on risk assessment of developing CVD after T2DM diagnosis. They found that personalized integrative care is associated with better care, compared to conventional care. Although it provides useful information to the field, there are some points that need to be mentioned.

Response: We greatly appreciate your recognition of the value our study brings to the field and your constructive feedback. Your insights are instrumental in guiding us to refine and enhance the quality of our research. We are committed to addressing the points you have raised and look forward to incorporating these suggestions to improve the clarity and impact of our findings.

Reviewer 1: In the abstract, the aim of the study should be stated at the end of the introduction/background.

Response: Thank you for your suggestion. We have revised the abstract to include the aim of the study at the end of the introduction. The updated Introduction of the Abstract now reads: “Type 2 diabetes mellitus (T2DM), a chronic metabolic disorder, significantly increases cardiovascular disease (CVD) risk. Integrative care (IC) offers a personalized health management approach, utilizing various interventions to mitigate this risk. However, the impact of IC on CVD risk in newly diagnosed T2Dm patients remains unclear. This study aims to assess the differences in CVD risk development within 120 months following a new diagnosis of T2DM, using real-world data from Bumrungrad International Hospital and Vitallife Scientific Wellness Center.”

Reviewer 1: The introduction section is rather long. Authors should focus on the main ideas related to the manuscript.

Response: Thank you for your valuable feedback regarding the length of the Introduction section. We have taken your suggestion into consideration and have revised this section to be more concise. In doing so, we focused on distilling the content to encompass only the main ideas directly relevant to our study. This revision ensures that the Introduction now more effectively sets the stage for our research, highlighting the key aspects of Integrative Care and its significance in the context of Type 2 Diabetes Mellitus and cardiovascular disease risk, without delving into less central details.

Reviewer 1: Figure captions are missed and should be added.

Response: Thank you for pointing out the omission of the figure captions in our manuscript. We appreciate your attention to detail. As per your suggestion, we have now added descriptive captions under each figure. These captions provide a brief explanation of the content and context of the figures, contributing to a better understanding and interpretation of the visual data presented in our study.

Reviewer 1: The first paragraph of the discussion should focus on the most eye-catching findings of the current study. Please summarize the most important findings there.

Response: Thank you for your valuable feedback. In line with your suggestion to emphasize the most significant findings in the opening paragraph of the Discussion section, we have revised it to directly highlight our primary discovery: Integrative Care (IC) substantially reduces the risk of cardiovascular disease (CVD) development in patients with type 2 diabetes mellitus (T2DM). This key result demonstrates the effectiveness of IC compared to conventional clinical care (CC), particularly in a long-term, 20-year cohort study involving patients at both Bumrungrad International Hospital (BIH) and VitalLife Scientific Wellness Center (VSWC). We believe that placing this crucial finding at the forefront of our discussion aligns with the significance of our study and addresses your concern for a focused and impactful opening.

Reviewer 1: It is better to add the survival table and number of patients at risk during different time intervals below the Kaplan-Meier survival plots.

Response: We have now included a survival table beneath each Kaplan-Meier plot, which provides detailed information about the number of patients at risk during various time intervals throughout the study. 

Reviewer 1: In Tables 3 and 4, it is highly suggested to add 95% CI to the HRs provided.

Response: We have updated both tables to include the 95% confidence intervals alongside each hazard ratio.

Reviewer 2: The study is well-written and centers around an intriguing subject. However, certain aspects of the methodology require further clarification.

Response: We are grateful for your positive remarks about our manuscript and appreciate your interest in our study’s subject matter. Your feedback highlighting the need for further clarification in our methodology is well-received. We understand the importance of clearly presenting our methodology to ensure the robustness and replicability of our research findings.

Reviewer 2: 1- The concept of integrative care (IC) lacks clarity, as the authors do not provide specific details regarding the precise components of IC intervention in their methodology. The authors neglected to provide a comprehensive account of the specific measures taken for the patient population and their corresponding utilization rate. The study primarily focuses on various aspects of IC quality, such as patient compliance.

Response: Thank you for highlighting the need for clarity regarding the Integrative Care (IC) components in our study. Your comment brings to light a common challenge in the field of personalized wellness care – the difficulty in providing a standardized description of IC interventions due to their inherently personalized nature.

In our study, we described the operational definition of IC as it is implemented at the BI-VitalLife Cohort, encompassing Collaborative Care Planning, Care Coordination and Communication, Continuity of Care, and a Patient-Centric Approach. This description is reflective of the personalized and adaptable nature of IC, where interventions are tailored to each patient’s unique needs, making standardization and detailed enumeration challenging.

This challenge is not unique to our study but is a characteristic feature of wellness centers that provide personalized care. The dynamic and individualized nature of IC means that the components can vary significantly from one patient to another, which can limit the ability to provide a detailed, uniform description applicable to all patients.

We understand that this aspect may affect the replicability of our study’s findings in different settings. However, we believe that our description provides a general framework of IC practices, which, despite their variability, share common principles and objectives in patient care.

In light of your feedback, we emphasize the need for future research to explore methodologies that can capture the essence of personalized IC interventions while allowing for the variability inherent in such approaches. This could contribute significantly to the standardization and clearer definition of IC in the field of wellness care.

We appreciate your insightful comments, which have helped us articulate the challenges faced in describing personalized IC interventions and have highlighted an important area for future research in this field.

Reviewer 2: 2- What was the author's rationale for considering the number of visits to VSWC as a proxy for compliance?

Response: Thank you for your question regarding the rationale behind using the number of visits to the VitalLife Scientific Wellness Center (VSWC) as a proxy for patient compliance in our study.

Our decision to consider the frequency of visits as a marker for compliance is based on the nature of Integrative Care (IC), which often involves a continuous and dynamic treatment regimen. In IC, adherence to medication and the regular adjustment of treatment plans are critical components. These adjustments are typically made during patient visits, based on their current health status and response to ongoing treatments.

Therefore, we hypothesized that a higher number of visits to VSWC would likely indicate a greater level of patient engagement and compliance with the prescribed IC regimen. This is because regular visits provide opportunities for patients to receive ongoing guidance, adjustments to their treatment plans, and necessary support, which are essential for effective IC.

Reviewer 2: 3- What is the author's rationale for excluding patients who had received an anti-diabetic agent, considering that they likely constitute a significant proportion of the diabetic population in any given community?

Response: The decision to exclude these patients was primarily based on the concern that the use of anti-diabetic agents could obscure the actual date of Type 2 Diabetes Mellitus (T2DM) diagnosis. Accurately identifying the onset of T2DM is crucial for our study, as it serves as the baseline from which we assess the risk of developing cardiovascular diseases (CVD) over time.

Patients already on anti-diabetic medication at the time of diagnosis could represent a more advanced stage of the disease or different management strategies that might not be directly comparable to those who were not on such medication. Including these patients could introduce a significant population bias, affecting the validity of our comparisons between individuals with T2DM receiving Integrative Care (IC) and those under conventional care.

We aimed to create a more homogenous study population to accurately assess the impact of IC on CVD risk in T2DM patients, which necessitated the exclusion of this particular subgroup. We appreciate your inquiry and hope this explanation clarifies our methodological choices.

Reviewer 2: 4- The relationship between cardiovascular disease (CVD) and diabetes mellitus (DM) is independent of which condition occurs first. However, it is certain that newly diagnosed DM can have a negative impact on the development and progression of previously diagnosed CVD. Considering this fact, the decision made by the authors to exclude patients who had been diagnosed with CVD prior to their type 2 diabetes mellitus (T2DM) diagnosis is debatable and requires clarification.

Response: Thank you for your insightful question regarding our decision to exclude patients with a pre-existing cardiovascular disease (CVD) diagnosis prior to their Type 2 Diabetes Mellitus (T2DM) diagnosis.

The primary focus of our study was to examine the development of CVD in patients following a new diagnosis of T2DM. This approach was chosen to more accurately assess the impact of Integrative Care (IC) on the progression of CVD in a relatively controlled diabetic population. Including patients who were diagnosed with CVD prior to T2DM would have introduced considerable variability in our study population, as these individuals might have had a different course of disease progression and management.

Moreover, when CVD is diagnosed before T2DM, it is challenging to ascertain whether the patient had undiagnosed T2DM at the time of their CVD diagnosis, particularly if the T2DM diagnosis occurred at another institution. This uncertainty could significantly confound our results, as we aimed to analyze the effect of IC from the onset of T2DM.

We acknowledge that this decision does limit the scope of our findings to a specific patient group. However, we believe that this approach was necessary to maintain the clarity and specificity of our study objectives. We appreciate your attention to this aspect of our methodology and hope this explanation provides the necessary clarification.

Reviewer 2: 5- If the objective of the study is to assess newly diagnosed patients, this should be clearly stated in the title, abstract, and methodology.

Response: Thank you for pointing out the need for clarity in our study’s focus on newly diagnosed patients. We recognize the importance of accurately reflecting the study’s scope in the title, abstract, and methodology to ensure that readers have a clear understanding of our research objectives from the outset. In response to your feedback, we have revised the title, abstract, and methodology sections of our manuscript to explicitly state that our study concentrates on patients newly diagnosed with Type 2 Diabetes Mellitus (T2DM). This revision will make it clear that our research assesses the development of cardiovascular diseases (CVD) post-T2DM diagnosis in a newly diagnosed patient population. We appreciate your guidance in enhancing the precision and transparency of our manuscript and believe that these adjustments will significantly improve the clarity of our study’s aim and scope.

Reviewer 2: 6- The authors fail to specify the adjusted variables employed and merely mention the imprecise term "significant laboratory variables." The request is for a detailed presentation of the "significant laboratory variables" and an explanation of why they were selected as confounding variables.

Response: We have revised our manuscript to clearly specify the laboratory variables that were adjusted for in our analysis. These include fasting blood glucose, HbA1c levels, lipid profile parameters (including LDL, HDL, and triglycerides), and renal function indicators (such as serum creatinine and estimated glomerular filtration rate). These variables were selected based on their established relevance in the progression and management of Type 2 Diabetes Mellitus (T2DM) and its associated cardiovascular risks. They are commonly used in clinical practice to monitor the health status of diabetic patients and are known to influence the development of cardiovascular diseases (CVD).

This selection was made to ensure that our analysis accounted for potential confounding factors that could impact the relationship between integrative care and the development of CVD in T2DM patients. By adjusting for these variables, we aimed to provide a more accurate and reliable assessment of the impact of integrative care on CVD risk in our study population.

Reviewer 2: 7- Using “increased and decreased amounts” and ranges to represent the results cause confusion for readers.

Response: We have revised the Results section of our manuscript. Instead of using generalized descriptive phrases, we have now included specific numerical values and statistical measures, such as exact percentages, mean values, standard deviations, and p-values, to clearly represent our findings. This change will allow for a more precise and unambiguous understanding of the data.

Reviewer 2: 8- In the section titled "Operational Definition of Integrative Care," the authors provide a list of IC goals as "Various components of integrated care," but they do not clearly define the specific intervention that was carried out.

Response: In our study, the nature of Integrative Care (IC) at the VitalLife Scientific Wellness Center is inherently personalized and dynamic, making it challenging to list specific interventions uniformly applied to all patients. IC interventions are tailored to each patient’s unique health profile, which includes a wide range of factors such as medical history, lifestyle, and specific health goals. This individualized approach results in a diverse array of interventions, differing significantly from one patient to another.

Given these constraints, it was not feasible for our study to track and report every specific intervention for each patient within the dataset. However, we aimed to convey the general scope and nature of IC through the listed components, such as collaborative care planning and patient-centric approaches.

We appreciate your feedback on this matter and acknowledge that this is a limitation of our study. We hope that the general outline provided offers some insight into the nature of IC as implemented in our cohort, even though specific intervention details could not be comprehensively included.

Reviewer 2: 9- Please provide a more explicit explanation of how the choice of a study duration of 120 months is connected to the inspiration derived from the ASCVD 10-year prediction and what is the significance of this particular time frame?

Response: Our decision to use a 120-month (10-year) timeframe for this study

---

## [Decision Letter · Decision Letter 1]

4 Apr 2024

Impact of Integrative Care on Cardiovascular Disease Risk in Newly Diagnosed Type 2 Diabetes Mellitus Patients: A BI-VitalLife Cohort Study.

PONE-D-23-25653R1

Dear Dr. Pongpirul,

We’re pleased to inform you that your manuscript has been judged scientifically suitable for publication and will be formally accepted for publication once it meets all outstanding technical requirements.

Kind regards,

Amirmohammad Khalaji

Academic Editor

PLOS ONE

Additional Editor Comments (optional):

Reviewers' comments:

Reviewer's Responses to Questions

**Comments to the Author**

1. If the authors have adequately addressed your comments raised in a previous round of review and you feel that this manuscript is now acceptable for publication, you may indicate that here to bypass the “Comments to the Author” section, enter your conflict of interest statement in the “Confidential to Editor” section, and submit your "Accept" recommendation.

Reviewer #1: All comments have been addressed

Reviewer #2: All comments have been addressed

2. Is the manuscript technically sound, and do the data support the conclusions?

Reviewer #1: Yes

Reviewer #2: Yes

3. Has the statistical analysis been performed appropriately and rigorously? 

Reviewer #1: Yes

Reviewer #2: Yes

4. Have the authors made all data underlying the findings in their manuscript fully available?

Reviewer #1: (No Response)

Reviewer #2: Yes

5. Is the manuscript presented in an intelligible fashion and written in standard English?

Reviewer #1: Yes

Reviewer #2: Yes

6. Review Comments to the Author

Reviewer #1: (No Response)

Reviewer #2: My concerns have been addressed in an acceptable manner. I consider the manuscript to be ready for publication.

7. PLOS authors have the option to publish the peer review history of their article (what does this mean?). If published, this will include your full peer review and any attached files.

Reviewer #1: No

Reviewer #2: No

---

## [Editor Report · Acceptance letter]

9 May 2024

PONE-D-23-25653R1 

PLOS ONE

Dear Dr. Pongpirul, 

I'm pleased to inform you that your manuscript has been deemed suitable for publication in PLOS ONE. Congratulations! Your manuscript is now being handed over to our production team.

Kind regards, 

on behalf of

Dr. Amirmohammad Khalaji 

Academic Editor

PLOS ONE